# Chemically Pretreated Densification of Juniper Wood for Potential Use in Osteosynthesis Bone Implants

**DOI:** 10.3390/jfb15100287

**Published:** 2024-09-28

**Authors:** Laura Andze, Vadims Nefjodovs, Martins Andzs, Marite Skute, Juris Zoldners, Martins Kapickis, Arita Dubnika, Janis Locs, Janis Vetra

**Affiliations:** 1Latvian State Institute of Wood Chemistry, Dzerbenes Street 27, LV-1006 Riga, Latvia; martins.andzs@kki.lv (M.A.); polarlapsa@inbox.lv (M.S.); jzoldn@inbox.lv (J.Z.); 2Department of Morphology, Institute of Anatomy and Anthropology, Riga Stradins University, Dzirciema Street 16, LV-1007 Riga, Latvia; vadims.nefjodovs@gmail.com (V.N.); janis.vetra@rsu.lv (J.V.); 3Microsurgery Centre of Latvia, Brivibas Gatve 410, LV-1024 Riga, Latvia; kapickis@sr.com; 4Institute of Biomaterials and Bioengineering, Faculty of Natural Science and Technology, Riga Technical University, Pulka Street 3, LV-1048 Riga, Latvia; arita.dubnika@rtu.lv (A.D.); janis.locs@rtu.lv (J.L.); 5Baltic Biomaterials Centre of Excellence, Headquarters at Riga Technical University, LV-1048 Riga, Latvia

**Keywords:** juniper wood, chemical pretreatment, kraft cooking, partial delignification, compressed solid wood, wood densification, wood bone implants, in vitro

## Abstract

The aim of the study was to perform treatment of juniper wood to obtain wood material with a density and mechanical properties comparable to bone, thus producing a potential material for use in osteosynthesis bone implants. In the first step, partial delignification of wood sample was obtained by Kraft cooking. The second step was extraction with ethanol, ethanol–water mixture, saline, and water to prevent the release of soluble compounds and increase biocompatibility. In the last step, the thermal densification at 100 °C for 24 h was implemented. The results obtained in the dry state are equivalent to the properties of bone. The swelling of chemically pre-treated densified wood was reduced compared to chemically untreated densified wood. Samples showed no cytotoxicity by in vitro cell assays. The results of the study showed that it is possible to obtain noncytotoxic wood samples with mechanical properties equivalent to bones by partial delignification, extraction, and densification. However, further research is needed to ensure the material’s shape stability, water resistance, and reduced swelling.

## 1. Introduction

There are two forms of surgical procedures used to heal bone disorders and injuries: bone grafting, which replaces missing, worn-out, or irreparably damaged bones, and osteosynthesis, which uses metal plates and screws to fix bone fractures. Material has to fulfill several requirements to be utilized as an osteosynthesis implant, including having mechanical qualities that exceed and are comparable to the host tissue and a good host–implant integration.

The top-performing material in osteosynthesis has been stainless steel, followed by titanium and cobalt-chromium alloys. These materials have become the industry norm for implants, used by millions of patients globally each year [1]. However, despite their success, these alloys are not without limitations. The trend towards non-metallic implants is rising due to various factors. Stainless steel and titanium are materials that are biocompatible, meaning they can coexist safely within the human body. They exhibit isotropic properties, meaning their mechanical characteristics are uniform regardless of the direction of force applied. Additionally, these materials are easily adaptable and can be molded into various configurations for medical applications. However, despite their favorable physical characteristics and ability to maintain their form within the biological environment, neither stainless steel nor titanium possesses the osteogenic properties that support and enhance bone regeneration and healing processes. The density of titanium alloys is typically up to three times higher than that of bones [2], making the metal stronger but potentially leading to increased bone fragility after healing. Hence, there is a risk of aseptic loosening of the metal implants as a potential complication up to 15 years post-surgery [3]. Repeated surgeries may be necessary to remove implants that have been rejected by the bone after healing. Studies have shown a risk of bio-corrosion in stainless steel and titanium alloys [4,5]. Furthermore, metal implants can pose challenges for certain medical tests and procedures [6,7]. Hence, alternative materials such as ceramics [8] like calcium phosphate, calcium carbonate, calcium sulphate, hydroxyapatite [9,10], and various polymers, as well as numerous copolymers, are currently undergoing testing [11,12]. Despite their excellent biocompatibility, these materials are often too brittle and do not possess the required mechanical strength for orthopedic applications.

The growing focus on environmental issues has prompted experts in the field of bone implant technology to delve into biomaterial research [13]. Recent studies have compared the ecological impact of various materials [13,14], explored the use of natural and bio-based polymers in bone regeneration [14,15,16], and investigated 3D printing as an eco-friendly option for bone implants [17]. Researchers have even looked into using materials sourced directly from nature, such as corals [14]. Despite these advancements, current materials do not meet the necessary mechanical standards for osteosynthesis. As a result, scientists continue to seek a strong and mechanically suitable bone implant made from renewable resources.

To the best of our knowledge, there are currently nine research papers discussing the potential use of solid wood as osteosynthesis of load-bearing bone implants, which have also been examined in a recent review article [18]. Six of these studies focused on untreated wood [19,20,21,22,23,24], while two looked at thermally treated wood [25,26] and only one study tested densified solid wood [27]. These studies investigated various types of wood such as birch, ash, willow, lime, bamboo, and juniper. Interestingly, only willow and lime showed a strong inflammatory reaction towards untreated wood, while birch, ash, and juniper exhibited good biocompatibility even without sterilization. Among the nine studies, only two looked at wood for osteosynthesis. Despite the positive findings, the inadequate mechanical properties of untreated wood may hinder its use for the osteosynthesis. By densifying the wood, its mechanical properties can be improved beyond those of bone, which is crucial for the osteosynthesis implants. Thus, the issues associated with metals, such as aseptic loosening and biocorrosion, could be avoided by substituting densified solid wood for metal implants while maintaining the biocompatibility of the wood with the bone [27].

While research on mechanical densification of wood dates back to the early 1900s, it was only recently that chemical or enzymatic pretreatment methods were developed [28,29]. In 2018, Song et al. introduced a new approach to wood densification, using a sulfite process to partially remove lignin from wood before hot-pressing [30]. Several studies have since tested this method of densification [31,32,33]. However, despite these advancements, the potential of densified wood as a biomaterial for osteosynthesis has not yet been explored. Wood shows potential as a material for osteosynthesis implants, but it also presents several challenges. These include anisotropic mechanical properties and dimensional changes in response to moisture, particularly in densified wood. Additionally, the presence of extractive substances in wood may lead to adverse reactions in bone cells. Both the benefits and drawbacks are discussed comprehensively in the review article [18].

The novelty of the research is confirmed by a previously unused type of chemical pretreatment (Kraft cooking) before densification and the evaluation of the obtained materials in the context of osteosynthesis bone implants, with mechanical properties, swelling and cytotoxicity. Even previous studies have not performed extraction after chemical pretreatment to prevent the release of cytotoxic compounds. Furthermore, we conduct mechanical properties testing following swelling in the wet condition to achieve results that are more representative of the human body’s environment.

## 2. Materials and Methods

### 2.1. Materials

Samples of solid juniper (Juniperus Communis) wood were gathered from a forest in Kegums, Vidzeme region, Latvia for the research. Sodium sulfide hydrate (≥60%, Sigma-Aldrich, Darmstadt, Germany), sodium hydroxide (>97%, Sigma-Aldrich, Darmstadt, Germany), ethanol (>95%) and deionized water were conducted for pretreatment of juniper solid wood sample.

### 2.2. Sample Preparation

Juniper logs, manually debarked, were left to air dry for 1 month before being cut into specimens measuring 90 mm × 15 mm × 15 mm (longitudinal × tangential × radial). The initial moisture content of the samples was 7.62%. The research methodology for densifying and characterizing juniper wood is depicted in Figure 1.

### 2.3. Chemical Treatment

Before treatment with chemicals, juniper wood specimens were immersed in a typical Kraft cooking solution made from 1.25 M NaOH and 0.25 M Na_2_S, commonly used in the pulping industry. The specimens were placed in individual autoclaves filled with the solution for 24 h. Subsequently, the autoclaves were heated to 165 °C in a glycerin bath and cooked at this temperature for various durations—removed immediately after reaching 165 °C (K0), after—10 min (K10), 20 min (K20), and 40 min (K40). The cooked specimens were then thoroughly washed with deionized water until no more coloring was observed, and, finally, stored in water.

The extraction process was as follows. To prevent the potential impact of soluble compounds on bone cells during in vitro assays, sample extraction was performed after cooking/before densification. Alcohol—water/alcohol (1:1)—and alcohol extraction was conducted for 14 h, 36 h, and 7 h, respectively, by simmering the sample in 200 mL of the appropriate solution for 1 h, straining the solution, and continuing the process until the solvent visually no longer changes color. Extraction was performed for only one sample after 10 min of Kraft cooking. The sample was named K10E. Samples’ abbreviations and corresponding treatments are summarized in Table 1.

### 2.4. Densification

After chemical treatment, juniper wood specimens in the radial direction (Figure 2A) underwent hot-pressing in a specially designed mold (Figure 3B) using a single-stage press LAP 40 (Gotfried Joos Maschinefabrik GmbH & Co., Pfalzgrafenweiler, Germany). The pressure applied was 5 MPa at 100 °C for 24 h, followed by an additional 12 h of interrupted heating. Control samples of dry (DD) and wet (WD) juniper wood were also hot pressed using the same method to compare the effects of chemical pretreatment. Immersion in hot (90 °C) water was used to prepare the WD sample before hot pressing. Three specimens of each sample type were produced for evaluation.

### 2.5. SEM

In order to analyze cross-section of the samples using scanning electron microscopy (SEM), a thin layer of gold plasma was applied to the samples using a K550X sputter coater (Emitech, South Petherton, UK). The samples were then examined using a Vega TC microscope (Tescan, Brno, Czech Republic), running software version 2.9.9.21.

### 2.6. Chemical Characterization

#### 2.6.1. Mass Loss

The method for determining mass loss (ML%) of samples after chemical pretreatment involved calculating the percentage of weight lost using Equation (1):(1)$${ML}_{\%}=\frac{M_{1}-M_{0}}{M_{0}}\cdot 100\%$$
where M0 and M1 represent the initial and final masses of the completely dried specimen before and after undergoing chemical pretreatment, respectively.

For chemical characterization, wood samples were ground using M20 mill (IKA-WERKE, Breisgau, Germany) before and after chemical pretreatment.

#### 2.6.2. Extractives

The ground samples were Soxhlet-extracted with acetone for 8 h to determine the amount of extractable components. This quantification was done gravimetrically using an ES 225SM-DR scale (Precisa, Zurich, Switzerland) after rotary vacuum evaporation with a PC3001 VARIO equipment (Green Vac, Düsseldorf, Germany). The results were expressed as a percentage of the initial wood sample mass, calculated using Equation (2)
(2)$${Ex}_{\%}=\frac{M_{2}-M_{1}}{M_{0}}\cdot 100\%$$
where, Ex%—the extractives in the wood sample, M—the absolutely dry sample mass, M1—the mass of the absolutely dry round flask, and M2—the mass of the absolutely dry round flask with sample extractives.

#### 2.6.3. Klason Lignin

Klason lignin was conducted under the TAPPI 222om-98 protocol.

#### 2.6.4. Chemical Composition

The HPLC (high-performance liquid chromatography) analysis was conducted by NREL/TP-510-42618 Laboratory Analytical Procedure for the determination of structural carbohydrates and lignin in biomass [34].

Hydrolysates were prepared and analyzed using a Shimadzu LC-20A HPLC system (Shimadzu, Tokyo, Japan) with a refractive index detector. Reference standards with a purity of ≥99.0% (Sigma Aldrich, Steinheim, Germany) were used for the analysis of glucose, cellobiose, arabinose, 2-furaldehyde, acetic acid, 5-HMF, levulinic acid, and formic acid by Shodex Sugar SH1821 column at 60 °C, with eluent 0.008 M H_2_SO_4_ (Sigma Aldrich, Germany) at a flow rate of 0.6 mL·min^−1^. Xylose, arabinose, galactose, and mannose were analyzed on Shodex Sugar SP0810 column at 80 °C, with deionized water as the mobile phase under a flow rate of 0.6 mL·min^−1^ using standards with a purity of ≥99.0% (Sigma Aldrich, Germany). Before injection, samples were neutralized to pH 5–7 with NaHCO_3_ and filtered through a 0.45 μm membrane filter.

#### 2.6.5. FTIR

Fourier transform infrared (FTIR) spectra of wood samples were obtained in KBr (IR grade, Sigma Aldrich, Darmstadt, Germany) pellets using Thermo Fisher Nicolet iS50 spectrometer (Waltham, MA, USA). The spectra were recorded within the range of 4000–450 cm^−1^, with a spectral resolution of 4 cm^−1^ and 32 scans. A pellet containing 2 mg of a ground sample and 200 mg of KBr was used for the measurements. The resulting spectra were then normalized to the highest absorption maxima.

### 2.7. Physical-Mechanical Properties

#### 2.7.1. Density

The density of each densified sample was measured after hot pressing and conditioning (25 °C; relative humidity 50%) and determined based on Equation (3):(3)$$\rho =\frac{M}{t\;\cdot \;w\;\cdot \;l}$$
where ρ (kg m^−3^) is the density and M (kg), t (m), w (m), and l (m) are the mass, length, width, and thickness of the conditioned samples, respectively.

#### 2.7.2. Swelling

The swelling test was performed in boiling saline (9% NaCl) at 100 °C for 2 h using samples with initial dimensions of 1.5 cm in length and 1.5 cm in width and corresponding sample thickness. After soaking, the swelling of the sample in volume % was calculated by measuring all dimensions based on Equation (4), and the density of the wood of the sample after swelling was calculated by Equation (4).
(4)$${Sw}_{\%}=\frac{V_{2}-V_{1}}{V_{1}}\cdot 100\%$$
where V_1_ (m^3^) is the initial volume of the sample and V_2_ (m^3^) is the final volume of the sample.

#### 2.7.3. Three-Point Bending

The densified juniper samples underwent testing for mechanical properties by the modulus of elasticity, more commonly used as Young’s modulus, and modulus of rupture, commonly known as strength, in a three-point bending test conducted on a ZWICK/Z100 (Ulm, Germany) universal testing machine following EN 310 (1993). The distance between the specimens was 70 mm. The three-point bending test was performed in the longitudinal direction as indicated in Figure 2D.

Three specimens of each sample type were analyzed to determine the average property values for strength and resistance. The length and width of all samples retained their original dimensions of 90 mm and 15 mm, respectively. The thickness of the control samples also stayed at the original measurement of 15 mm. In contrast, the thickness of the compacted materials varied according to the results obtained after the densification process (refer to Table 2).

### 2.8. In Vitro Analysis

The cytotoxicity of the sample was assessed using the NIH 3T3 mouse fibroblasts cell line. Cells were pre-seeded in 96-well plates in a density of 5 × 10^3^ cells in 100 μL cell medium (89% Dulbecco’s Modified Eagle Medium (DMEM), supplemented with 10% (*v*/*v*) calf serum and 1% penicillin/streptomycin (P/S)) and incubated (New Brunswick™ S41i CO_2_ Incubator Shaker, Eppendorf, Hamburg, Germany) for 24 h to allow them to attach. Then, the medium was replaced with 100 μL of sample extract dilutions.

To obtain the sample extracts, the sample was placed in 5 mL of fresh cell medium. After 24 and 48 h, all the solution was collected from the samples and replaced with an additional 5 mL of fresh cell medium. The collected solution was then filtered through a 0.2 μm syringe filter and, subsequently, used at 100% for dilution with fresh medium. The extract dilution—1:1—was chosen for this experiment. Untreated cells were used as positive control. The negative control was cell medium with 10% dimethyl sulfoxide (DMSO, Sigma-Aldrich). After incubating the cells with the sample extract and its dilution in a humidified atmosphere with 5% CO_2_ and 90% humidity at 37 °C for 24 h and 48 h, all medium in the 96-well plates were replaced with 120 μL CellTiter-Blue (CTB) prepared in cell medium to each well. The plates were put back into the cell incubator for 2 h before optical density was measured by Infinite M Nano microplate reader (Tecan, Männedorf, Switzerland) at 590 nm.

### 2.9. Statistic

The statistical analysis was carried out using the data analysis tool in SPSS 17.0. Mean values (MV), standard deviation and standard error (SE) were determined from three parallel measurements. The data underwent testing for normality and homogeneity of variances using Levene’s test. When the data met the criteria of being normally distributed, a one-way ANOVA test was conducted. For comparisons between two independent samples, Student’s *t*-test was employed. All data were reported in the format of MV ± SE. The significance level for all statistical tests was set at α = 0.05.

## 3. Results and Discussion

Some of the research data (density and mechanical properties of the samples—untreated, DD, WD, K0, K20, and K40) have been previously published [27]; however, they have been re-included in this publication to demonstrate the integration of the newest data with the existing dataset and to elaborate on prior results with additional information on chemical analysis, FTIR, SEM, swelling properties, in vitro analysis, and mechanical properties of extracted sample.

### 3.1. Visual Appearance and SEM

The visual representation of all densified juniper wood samples can be seen in previous publication [27]. The juniper wood K10E sample’s appearance has changed after chemical treatment and densification displaying a dark brown color (Figure 3). Changes are linked to the Kraft cooking method, which causes the breakdown of lignin and hemicelluloses in the wood cell walls and the extraction of extractives [35,36]. Chemical changes are described in the Table 1. The change in color in chemically pretreated and densified juniper wood may be attributed to the chemical breakdown of hemicelluloses and lignin, leading to the formation of new chromophoric groups. Additionally, the thermochromatism of the chemicals applied on the wood surface may also play a role [37].

SEM analysis was performed to demonstrated wood structural changes in the cell level after partial delignification and densification. The comparison of microstructures of untreated (A) and densified (B) wood samples in the crossfiber direction at equal magnification is depicted in Figure 4.

An untreated juniper wood sample (Figure 4A) displays cylindrical tracheids with a lumen diameter ranging from 100 to 200 µm. These tracheids are organized in a radial pattern, typical of softwood. The tracheid lumens in the transverse section of the densified wood were nearly imperceptible (Figure 4B). Observing the changes in the microstructure of juniper wood suggests that the wood tracheid underwent sufficient densification and recombination as a result of the chemical treatment and hot pressing. It is likely that the process of partial delignification only softened and plasticized lignin, as well as partially separated readily available lignin in the middle lamella, which weakens the integrity of the cell walls, causing distortions in the lumen structure [37,38].

Similar microstructure changes in wood have also been assessed in the works of other authors, using deep eutectic solvents [39], sulfite [30,31,40], and acid or alkyl pretreatments [38], as well as steam or hydrothermal treatment [41].

### 3.2. Chemical Characterization

The Kraft process, a leading pulping technology globally, involves treating lignocellulosic feedstock in an alkaline solution containing hydroxide, sulfide, and bisulfide ions. This causes the dissolution of lignin and some hemicelluloses [42]. As it can be seen in Table 3, during chemical pretreatment, mass losses of 21–32% occur, primarily due to hemicelluloses dissolution and lignin breakdown. Hemicelluloses content decreases by 7–12% depending on processing severity. Lignin content in juniper wood reduced from 33% in untreated samples to 17% in K20 and K40 samples due to lignin depolymerization during Kraft cooking.

A slight decrease in extractives and inorganics was also observed.

### 3.3. FTIR

For studying the effects of chemical pretreatment on juniper wood polymers in the cell wall, we conducted an analysis by comparing FTIR spectra of untreated and treated samples (Figure 5). The peaks at 3378 cm^−1^ and 2900 cm^−1^ are associated with the stretching vibrations of O–H and C–H in cellulose, hemicellulose, and lignin [43,44]. These distinctive bands were also observed in both untreated and densified wood samples during analysis. The disappearance of the absorption band at 1729 cm^−1^, which is associated with the C=O stretching vibration in unconjugated ketones, is a characteristic feature of hemicellulose [31,37,40]. This band was no longer present in the densified wood due to deacetylation in the hemicellulose, demonstrating that the hemicellulose had dissolved in the NaOH/Na_2_S solution. Untreated juniper wood showed absorption peaks in the 1588–1560 cm^−1^ region of the spectrum as an absorption shoulder, and similar absorption shoulders were still present after chemical treatment. This indicates that there has been no alteration in the polar functional groups attached to the benzene ring in lignin. In contrast, Shi et al. found changes in these functional groups when using chemical treatments involving NaOH/NaSO_3_ [31] and H_2_SO_4_ [37]. Additionally, there was no evidence of a new peak at 1714 cm^−1^ [37], typically indicating pseudo-lignin formation resulting from significant lignin and hemicellulose depolymerization and repolymerization. This demonstrates that the chosen chemical treatment only partially breaks down the lignin and hemicellulose in the cell walls, but rather softens and makes them more plasticize.

The sharp band at 1507 cm^−1^ at was identified as resulting from the vibrational absorption of the benzene ring structure. In chemically treated wood, the absorption band remains present, but its intensity decreases over time as the treatment progresses. This suggests a reduction in lignin content, aligning with the findings of the chemical analysis detailed in previous section. The decrease of the absorption peaks at the wavelengths of 1261 cm^−1^ and 1246 cm^−1^, are also related to the decrease of the lignin content, representing guaiacol and syringyl groups, respectively [31,37,44].

### 3.4. Swelling

Swelling and shrinking are two of the major disadvantages of wood as a material for use in environments with variable moisture content. Considering that muscle tissue consists of 76% water [45], assessing the swelling of wood implants in an aqueous environment is crucial. The most precise method involves measuring the material’s swelling in body fluid at 37 °C for an extended period. It takes 2–3 months to reach maximum swelling in water at room temperature [46]. To expedite the process and reach the maximum swelling value, swelling was evaluated by boiling the implant material in saline for two hours, as water is known to be the primary factor influencing swelling [46].

Table 4 shows the swelling results for untreated and densified juniper wood samples. Untreated juniper wood showed a minor swelling of 13%, while chemically untreated dry densified juniper wood (DD) exhibited a 22% of swelling. The most significant swelling was observed in the chemically untreated wet densified wood sample (WD), which fully recovered to its original dimensions and even surpassed them. Similar results were also obtained in the works of other authors [47,48,49].

Table 4 illustrates the pronounced anisotropy of wood, indicating that swelling primarily occurs in the radial direction—along the growth rings—while the longitudinal dimension remains unaffected. Given the anisotropic characteristics of wood, densification was executed in the radial direction, as this is where the material is most susceptible to physical, mechanical, and chemical influences. Observations reveal that densified wood exhibits the most significant swelling in the same radial direction as the densification process. This suggests that the densification did not establish stable, chemically irreversible bonds sufficient to prevent the material from reverting to its initial form. Future investigations will focus on potential methods such as chemical bonding and/or impregnation to enhance the dimensional stability of the wood during densification and to mitigate additional swelling.

Chemical pretreatment reduced swelling by 15–30% and provided better shape stability. There is no statistically significant variation observed among the chemically treated samples; however, a significant difference is found between the chemically treated compacted samples and the WD sample. To the best of our knowledge, there have been no studies by other authors on the swelling of densified chemically pretreated samples in aqueous media, but there are studies on swelling in different air humidity [50].

Laine et al. and Welzbacher et al. conducted research on the set-recovery in water conditions after densification and following thermal or oil treatment, respectively [47,48]. However, it should be noted that these studies did not perform a chemical pretreatment that reduces the cell density of wood prior to densification. In our study, the calculation of set-recovery may not be accurate due to the variations in wood structure between untreated and chemically treated samples.

### 3.5. Density

In our research, Kraft cooking pretreatment of wood material led to comparable density results (Figure 6.) to those found in studies using sulphite cooking [30,31,32,40] or deep eutectic solvent [39] pretreatment conducted by other researchers. Pretreatment with steam [41,51], acid, or alkyl [38] pretreatment in the work of other authors showed less increase in density for wood samples.

As can be seen in Figure 6a, chemical pretreatment of juniper wood and subsequent densification increased the density of the samples by 2.1–2.3 times depending on the treatment time. For sample K10E, the density increased by 2.5 times, reaching 1300 kg m^−3^, approaching the theoretical maximum density for wood of 1500 kg m^−3^. This upper limit is due to the density of the wood cell wall, which is typically around 1500 kg m^−3^ and remains relatively constant [54]. The densified juniper wood has a density comparable to that of cortical bone [18,27,52,53], making it a strong candidate for use in bone implant applications. Although the absolute values of the density values of the modified wood samples differ, they do not have a statistically significant difference between each other and also with the K10E sample. Even so, no statistically significant difference was observed between the chemically treated densified samples and the DW sample. In their research, Song et. al. found that reducing the lignin content in wood leads to higher density [30], although our own study did not replicate this result. Despite the fact that lignin has the lowest density (1350 kg m^−3^) within wood cells, with more delignification potentially boosting wood density. It is important to note that the Kraft process not only removes lignin but also hemicelluloses, which actually have a higher density (1500–1800 kg m^−3^) [54]. This could explain the lack of significant differences in wood density across various chemical treatment durations.

Although there is a statistically significant difference of swelling in water between the chemically treated and untreated densified juniper wood samples, the alteration in the wood’s microstructure due to chemical treatment still remains a factor that cannot be overlooked. In order to compare the swelling levels of untreated and treated samples the results were analyzed in terms of wood density post-swelling (Figure 6b).

Upon swelling, the WD sample returned to the original density of the sample, with no statistically significant difference from the densities of the DD and untreated samples. The density of the WD sample dropped by half while swelling, which is equal to the density increase during densification. The chemically treated densified samples showed a 35–45% decrease in density during swelling, maintaining a density 30–35% higher than the untreated sample before swelling. This indicates microstructural changes in wood during chemical treatment and densification. Wood cell walls have softened, lignin has become more plastic, and new links are formed in the microstructure of wood during thermal densification. There is no statistically significant difference between post-swelling densities of chemically treated densified juniper wood samples.

### 3.6. Mechanical Properties

The bending properties of juniper wood samples in dry state and after swelling are presented in Figure 7a,b and Figure 8a,b, accordingly. The dry control sample did not show adequate densification, but wet densification led to a notable increase in strength, as shown in Figure 7a. Therefore, the water content in the sample contributes to the formation of hydrogen bonds during the pressing stage, enhancing the densification process and resulting in higher density and improved bending properties.

It is important to note that the hydrogen bonds formed in the WD sample are not stable in a water environment, as illustrated in Figure 7b. The strength of all control samples (Untreated, DD, and WD) after swelling in saline does not show a statistically significant difference among them and is approximately 40% of the strength of untreated wood in a dry state, which aligns with the findings reported in the literature [55,56].

Chemical pretreatment and further densification of juniper wood improved strength in the dry state, with the sample treated for 10 min achieving the highest strength value. The chemical pretreatment of juniper wood has resulted in the partial degradation of lignin and hemicelluloses, leading to the formation of new covalent bonds that further enhance all identified properties. However, no statistically significant difference was observed between the chemically pretreated samples with each other as well as the wet-densified sample (WD). The chemically treated sample extracted pre-densification (K10E) demonstrated a statistically significant increase in strength value compared to the other samples. The combination of chemical pretreatment and densification resulted in a 70–90% increase (160–180 MPa) in strength value reaching a value 40% higher than bone, with the addition of extraction leading to a 2.3 times (215 MPa) greater increase in strength compared to the untreated sample.

In the studies of other authors, slightly lower strength absolute values (150 MPa) were achieved on densified poplar by hydrothermal densification [41] or NaOH pretreatment [38], but sulphite cooking pretreatment with subsequent impregnation with sodium silicate increased the poplar strength to 300 MPa [57]. For softwood, the strength value after densification with sulphite cooking pretreatment reached 220 MPa (*Abies*) [31], while deep eutectic solvent pretreatment increased the strength value to 180 MPa (*Pinus*) [39].

Figure 7b illustrates that chemically pretreated densified samples, after being swollen in boiling saline, exhibit a strength increase of 50–60% compared to the control sample, which is statistically significant. However, the chemical bonding formed during the densification process is minimal and does not sufficiently enhance the mechanical properties of the material under wet conditions. This indicates that further investigation is required, potentially involving impregnation or the enhancement of chemical bonds during densification, to minimize sample swelling and improve mechanical strength.

The impact of dry densification (DD) on the Young’s modulus value was not found to be statistically significant, whereas wet densification (WD) led to a 4.2-fold increase in the elasticity of untreated juniper (Figure 8a). Chemical pretreatment proved to be effective in improving the flexibility of juniper wood post-densification, resulting in a 6–7.2-fold increase in the Young’s modulus value compared to untreated samples. The results obtained are higher than for bone. The chemically treated samples have no statistically significant difference between each other and compared to the extracted sample (K10E).

In other authors’ studies on poplar densification by various treatments, higher Young’s modulus values (22 GPa) were achieved [38,39,41], but the increase is equivalent, because the Young’s modulus of untreated poplar is 4–5 GPa, while the Young’s modulus of juniper is 2 GPa. The cross-linked cellulose microfibrils, along with lignin and hemicelluloses, serve as a structural framework in wood cell wall. It has been observed that the elasticity of wood is negatively correlated with the lignin content present [31,32]. The high lignin content (34%) found in juniper wood may explain the lower modulus of elasticity seen in untreated juniper wood samples [58,59].

After boiling in saline, the Young’s modulus of the chemically pretreated densified samples decreased for times (Figure 8b). However, this value remains three times higher than that of the control samples (Untreated, DD, and WD). For K0 and K10, the elasticity under wet conditions is similar to the Young’s modulus of bone. This indicates that chemical pretreatment followed by densification enhances the mechanical properties of wood materials. Nevertheless, further research is necessary to improve chemical bonding and ensure dimensional stability in moist environments, which is essential for exploring these materials’ potential use in osteosynthesis implants.

### 3.7. In Vitro Analysis

In the conditional medium, untreated juniper wood samples (untreated, DD, and DW) and chemically pretreated densified samples (K0, K10, K20, and K40) both showed pH changes that excluded the indirect in vitro test. Due to this, in vitro tests on NIH 3T3 mouse fibroblasts cells were conducted on the sample K10E that underwent extraction to eliminate water- and alcohol-soluble compounds before densification, in order to avoid potential cytotoxic reactions.

The data from the in vitro cell studies indicated that lower relative metabolic activity is observed after 24 h in the conditioned medium—pure extract from K10E sample (see Figure 9). Nevertheless, the average value of cell viability was over 80%, corresponding to the non-cytotoxic sample according to the ISO 10993-5 standard [60] for medical device evaluation. At 50% or greater dilution, there is no statistical difference in relative metabolic activity between the positive control and evaluated sample. No cytotoxic effect was observed in the condition medium at any dilution after 48 h. The utilization of wood as a bone implant material was mainly explored towards the end of the 20th and beginning of 21st centuries. Publications from that era detail in vivo studies that indicate birch, ash, and juniper exhibit good biocompatibility properties [19,20,21,22,24]. On the other hand, lime and willow triggered acute inflammatory responses [23]. Moreover, untreated bamboo displayed cytotoxicity in vitro [61]. The in vitro inflammatory response is influenced by the soluble components of wood (extractives). While Gross et al. conducted in vivo experiments with untreated juniper that showed no toxic reactions [24], our own study found it impractical to conduct in vitro tests on untreated juniper. This suggests that, in certain instances, a material may demonstrate good biocompatibility in vivo analyses despite yielding negative in vitro results.

## 4. Conclusions

The findings of the study indicated that chemical pretreatment has a notable effect on enhancing the mechanical properties of densified wood, suggesting its potential applicability in osteosynthesis implants using densified wood. The results obtained in the study shows equivale mechanical properties as of bone, increasing density by 2.5 times, but strength and Young’s modulus by 2.3 and 7.2 times, respectively. Although the swelling is still relatively large, the swelling of chemically pre-treated densified wood was reduced by 15–30% compared to chemically untreated densified wood.

Additionally, it should be noted that the mechanical properties of chemically pretreated densified wood in its wet state do not meet the requirements for use as osteosynthesis implants.

The relative metabolic activity of the NIH 3T3 mouse fibroblasts suggests that the samples exhibit no significant cytotoxic effects at any of the tested dilutions.

The results of the study showed that it is possible to obtain nontoxic wood samples with mechanical properties equivalent to bones by partial delignification, extraction, impregnation and densification in dry state. However, it can be concluded that additional research is needed to enhance the stability of densified wood, the integrity of chemical bonds, and its mechanical strength in wet conditions, particularly in relation to human tissue. This research is essential for evaluating the material’s potential for application in osteosynthesis bone implants.

## Figures and Tables

**Figure 1 jfb-15-00287-f001:**
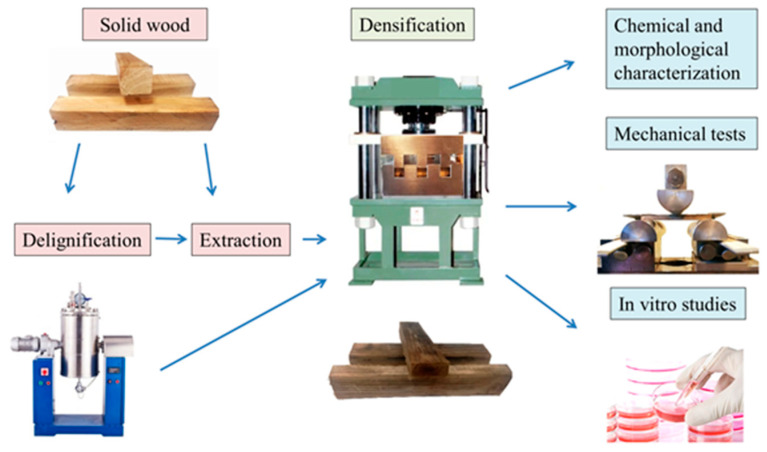
Schematic illustration of the obtained research.

**Figure 2 jfb-15-00287-f002:**
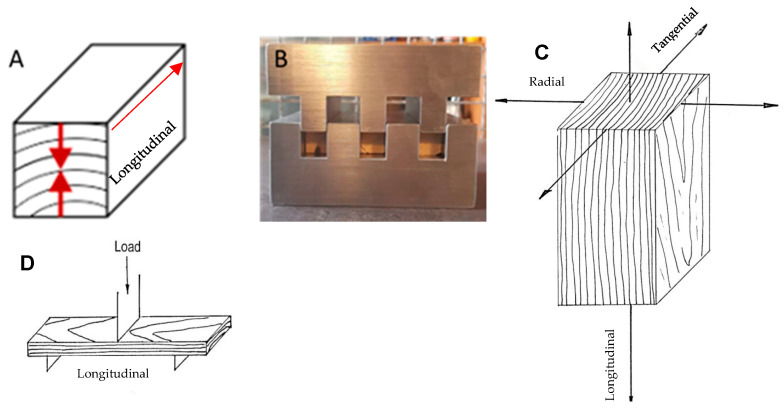
Densification. (**A**)—densification direction (radial). (**B**)—densification mold. (**C**)—anisotropy of the wood sample. (**D**)—direction of 3-point bending measurement.

**Figure 3 jfb-15-00287-f003:**
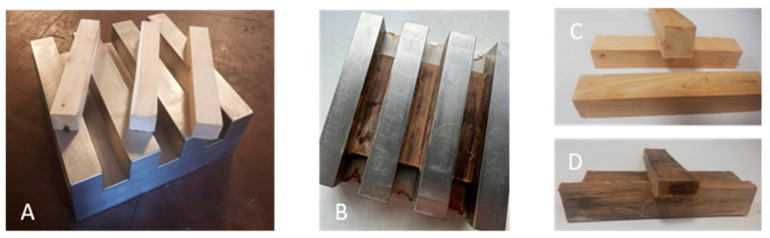
Visual evaluation of untreated sample (**C**) with press mold (**A**) and K10E sample (**D**) with press mold (**B**) after densification.

**Figure 4 jfb-15-00287-f004:**
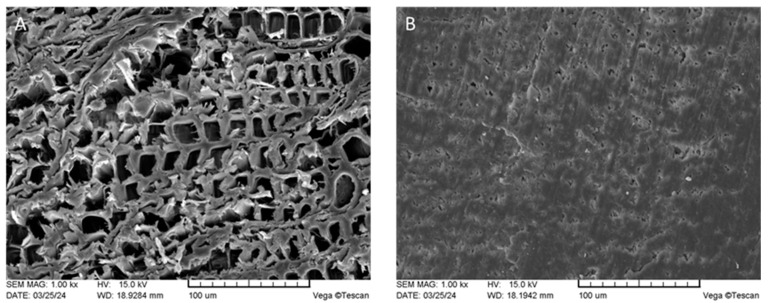
SEM images of untreated (**A**) and densified K10E (**B**) sample at 1000 magnification.

**Figure 5 jfb-15-00287-f005:**
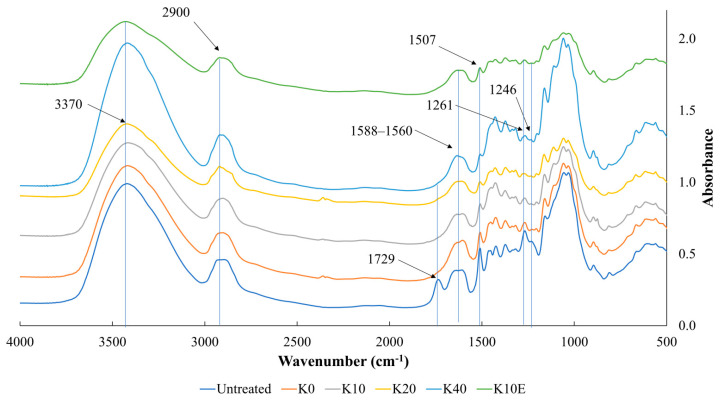
FTIR spectrum of untreated chemically treated sample.

**Figure 6 jfb-15-00287-f006:**
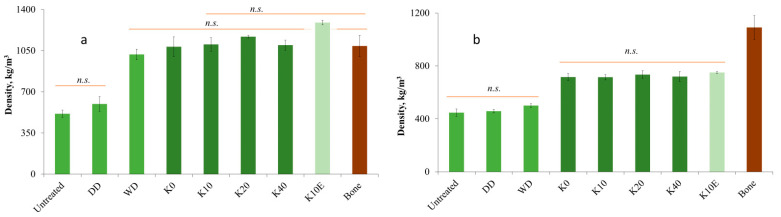
Density of untreated and densified juniper wood vs. cortical bone (**a**) in dry state and (**b**) after swelling in boiling saline [52,53]. n.s.—not statistically significantly different.

**Figure 7 jfb-15-00287-f007:**
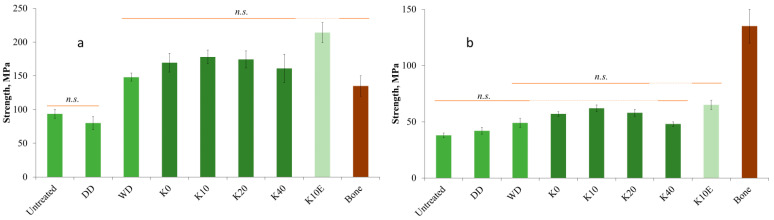
Strength of untreated and densified juniper wood vs. bone (**a**) in wet state and (**b**) after swelling in boiling saline [52]. n.s.—not statistically significantly different.

**Figure 8 jfb-15-00287-f008:**
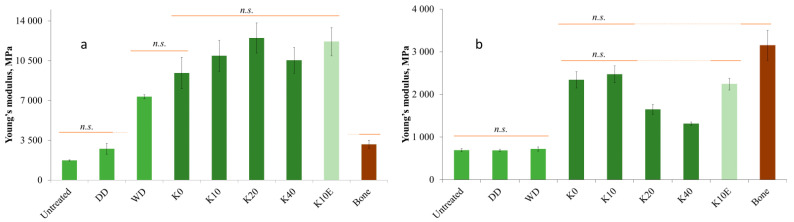
Young’s modulus of untreated and densified juniper wood vs. bone (**a**) in wet state and (**b**) after swelling in boiling saline [52]. n.s.—not statistically significantly different.

**Figure 9 jfb-15-00287-f009:**
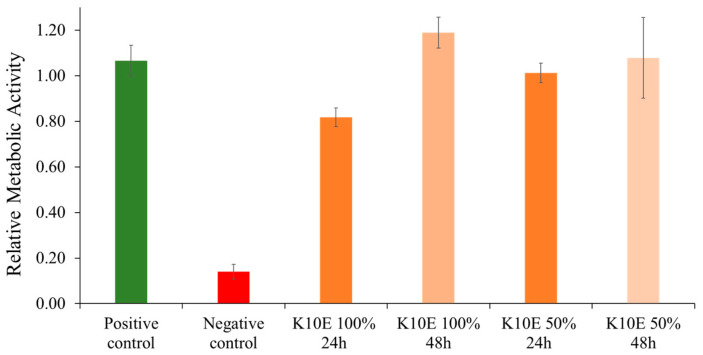
Relative metabolic activity of densified, chemically pretreated, and extracted juniper wood sample (K10E) assessed by CTB assay.

**Table 1 jfb-15-00287-t001:** Samples’ treatments and abbreviations.

Sample Abbreviation	Untreated	DD	WD	K0	K10	K20	K40	K10E
Cooking time (min) after reaching 165 °C	-	-	-	0	10	20	40	10
Treatment before densification	-	Dry	Soaked in water (wet)	Washed and soaked in water				Extracted by ethanol and water

**Table 2 jfb-15-00287-t002:** Thickness of sample for three-point bending test in dry and wet (after swelling) state.

Sample	Untreated	DD	WD	K0	K10	K20	K40	K10E
thickness, mm (dry)	14.93 ± 0.07	13.89 ± 0.06	8.05 ± 0.03	4.83 ± 0.02	4.81 ± 0.07	4.26 ± 0.04	3.82 ± 0.05	3.99 ± 0.04
thickness, mm (wet)	15.27 ± 0.11	15.20 ± 0.10	15.10 ± 0.11	8.07 ± 0.12	7.5 ± 0.3	8.07 ± 0.08	6.43 ± 0.13	5.70 ± 0.10

**Table 3 jfb-15-00287-t003:** Chemical characterization of wood samples before and after treatment.

	Weight Loss, %	Cellulose, %	Hemicelluloses, %	Lignin, %	Extracts, %	Inorganic Compounds, %	Other Components, %
Untreated	0	38.0 ± 2.1	21.1 ± 1.9	33.1 ± 0.5	4.8 ± 0.3	0.060 ± 0.002	2.94
0	21.7 ± 1.8	38.6 ± 0.5	14.4 ± 0.5	20.6 ± 0.1	3.3 ± 0.5	0.050 ± 0.003	1.35
K10	25.9 ± 2.0	38.3 ± 0.9	12.7 ± 1.2	18.2 ± 0.2	3.4 ± 0.2	0.041 ± 0.001	1.459
K20	27.2 ± 0.8	38.1 ± 0.9	12.2 ± 0.5	17.8 ± 0.5	3.2 ± 0.3	0.042 ± 0.005	1.458
K40	32.2 ± 1.2	36.4 ± 1.2	9.1 ± 1.3	17.3 ± 0.7	3.5 ± 0.7	0.040 ± 0.007	1.46

**Table 4 jfb-15-00287-t004:** Untreated and densified sample swelling in boiling saline.

Sample Swelling	Untreated	DD	WD	K0	K10	K20	K40	K10E
Volume, %	13 ± 6	22 ± 6	52 ± 3	34 ± 8	32 ± 4	44 ± 2	38 ± 4	44 ± 1
Longitudial, %	0.08 ± 0.02	0.11 ± 0.03	0.13 ± 0.09	0.09 ± 0.04	0.2 ± 0.1	0.10 ± 0.04	0.17 ± 0.01	0.19 ± 0.11
Tangential (width), %	5.5 ± 0.8	4.0 ± 0.6	4.9 ± 0.3	4.1 ± 0.7	4.8 ± 0.9	3.7 ± 0.1	4.0 ± 0.7	3.8 ± 0.5
Radial (thickness), %	20 ± 3	34 ± 5	67 ± 4	49 ± 5	57 ± 5	52 ± 2	60 ± 4	55 ± 3

## Data Availability

The original contributions presented in the study are included in the article, further inquiries can be directed to the corresponding author.

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
