# Peer review of "Chemically Pretreated Densification of Juniper Wood for Potential Use in Osteosynthesis Bone Implants"

_jfb, 2024, doi:10.3390/jfb15100287_

Round 1

Reviewer 1 Report

Comments and Suggestions for Authors

The manuscript "Chemically pretreated densification of juniper wood for potential use in osteosynthesis bone implants" presents treatment of juniper wood to obtain wood material with a density and mechanical properties comparable to bone thus producing a potential biomaterial with no cytotoxycity.

Some notes:

1) lines 50-51: please re-write the sentence. Indeed calcium phosphate, calcium carbonate, calcium sulphate, hydroxyapatite are among ceramic materials.

2) lines 64 and following: I suggest to refine the search. See for example DOI https://doi.org/10.1039/B900333A

3) Figure 5-please add vertical lines in order to allow direct comparison of bands positions among spectra

4) line 315 and following- This comment is related to figure 6

5) paragraph 3.5: I suggest to test swelling behavior during time, so to check if the measured values are a correct evaluation of the maximum swelling that the material can reach (plateau of the swellling curve - % vs time).

Author Response

Thank you for your suggestions for improving the manuscript.

The manuscript "Chemically pretreated densification of juniper wood for potential use in osteosynthesis bone implants" presents treatment of juniper wood to obtain wood material with a density and mechanical properties comparable to bone thus producing a potential biomaterial with no cytotoxycity.

Some notes:

  • lines 50-51: please re-write the sentence. Indeed calcium phosphate, calcium carbonate, calcium sulphate, hydroxyapatite are among ceramic materials.

Sentence re-writtened

2) lines 64 and following: I suggest to refine the search. See for example DOI https://doi.org/10.1039/B900333A

Thank you for the suggestion, our team has familiarized itself with a very wide volume of literature on the use of wood in bone implants, including this publication, while preparing the review article "Wood as Possible Renewable Material for Bone Implants—Literature Review" in the same journal. The sentence in line 64 was rephrased - there are only 9 articles that discuss solid wood as a potential ostesynthesis or other type of load-bearing implant, rather than a bone substitute. Your proposed article looks at the use of wood as a biomimic material and when the implant material is obtained, only the structure with C (carbon) is left of the wood.

  • Figure 5-please add vertical lines in order to allow direct comparison of bands positions among spectra

Vertical lines added

  • line 315 and following- This comment is related to figure 6

Corrected as per your suggestion.

  • paragraph 3.5: I suggest to test swelling behavior during time, so to check if the measured values are a correct evaluation of the maximum swelling that the material can reach (plateau of the swellling curve - % vs time).

Thank you for your suggestion. I completely concur that to achieve a more precise modeling of swelling, testing over an extended period is essential, with results represented as % swelling over time. To accomplish this, we would need to implement a prolonged testing process that mimics the moisture conditions found in the human body (exposed to body fluid) at a temperature of 36.6-37.0 degrees, which would span several months.

Given the extensive time commitment this involves, we opted for a preliminary study that utilizes a more extreme water treatment for the wood: boiling it for 2 hours. This method is known to cause complete swelling of the wood. While it's true that such extreme conditions are not reflective of what occurs in the human body, we believe this approach is sufficient for initial investigations.

Moving forward, we plan to conduct longer-term swelling tests, particularly in scenarios where special treatments such as impregnation are employed to reduce the swelling of the samples.

Reviewer 2 Report

Comments and Suggestions for Authors

See attached pdf file.

Comments on the Quality of English Language

Ok to me. Many typographical errors though. I have mentioned this in the review. 

Author Response

Thanks for the great suggestions to improve the publication. They were valuable and thought-provoking and helped to understand the direction of future research.

In response to your questions and suggestions:

  1. In order to improve the quality of the pattern, additional studies were carried out, re-creating samples and performing mechanical three-point bending results after swelling (boiling). To achieve a more precise modeling of swelling, testing over an extended period is essential, with results represented as % swelling over time. To accomplish this, we would need to implement a prolonged testing process that mimics the moisture conditions found in the human body (exposed to body fluid) at a temperature of 36.6-37.0 degrees, which would span several months. Given the extensive time commitment this involves, we opted for a preliminary study that utilizes a more extreme water treatment for the wood: boiling it for 2 hours. This method is known to cause complete swelling of the wood. While it's true that such extreme conditions are not reflective of what occurs in the human body, we believe this approach is sufficient for initial investigations. Moving forward, we plan to conduct longer-term swelling tests, particularly in scenarios where special treatments such as impregnation are employed to reduce the swelling of the samples
  2. Our goal is to create an implant for bone fracture osteosynthesis. The goal of this procedure is to restore the anatomical shape and stability of the bone. These properties are provided by the cortex of the bone. Thus, the trabecular part of the bone is insignificant from an orthopedic point of view. In the graphs, the density of the cortical bone was kept. Mechanical properties include the properties of the whole bone together, obtained by other authors using three-point bending measurement.
  3. The introduction has been slightly modified to include information about the disadvantages of wood and the advantages of metal to make the introduction more neutral.
  4. In the article, swelling results are now given both in volume and in each of the dimensions. It can be seen that the swelling occurs in principle only in the compacted part (Radial direction), and very little in the tangential direction. Both bone and wood properties are measured longitudinally. This is now described in the manuscript.
  5. The aim is to create an implant that provides bone fracture fixation until the bone heals. According to the bone fracture healing pathophysiology there 2 ways of bone healing – direct and indirect. For most of the fractures, indirect healing, i.e. through callus formation, is preferred. For this path of healing micromovements of the fractured parts are required. Thus, the mechanical requirements of the implant are lower, especially compared to metallic implants. Clinically mechanical loading is forbidden for any fractures for at least 4 weeks. The implant at best has to provide mechanical strength for limb mild movements without gravity load. To find out the necessary values of the mechanical properties, further studies with biomechanical models will be conducted in the near future. In the specific situation, we look at the obtained results so that the properties of the obtained material are equal to or greater than the properties of the bone.
  6. Section removed.
  7. Names of mechanical properties changed from MOR and MOE to Strength and Young's modulus.
  8. The authors apologize for the situation caused by careless errors. The manuscript has been reviewed and the errors mentioned have been corrected.
  9. Material codes are placed in the table.
  10. Specimen dimensions and parameters of the three-point bending method are summarized in the Materials and Methods section.
  11. Comparison with bone density was seen in Figure 6, but is now also added to the post-swelling images.
  12. Conclusions improved.

Round 2

Reviewer 2 Report

Comments and Suggestions for Authors

I think that the authors have addressed all the points raised in the review in the revised version of the manuscript. However, it was not easy to find the improvements, since that it was only written that changes have been made in the response, not where and not reciting the changes, which is normally done and would save a lot of time for reviewers. 

In language does not seem to have been properly revised in the new version: Noun 'Longitude" used instead of adjective 'Longitudinal'. "Young's modulus index" sounds very strange. It is called "Young's modulus" only. It is spelled "Thickness", not "tichness". These were found after a quick read-through. I would expect that there are more errors, which can risk to irritate some readers. 

Comments on the Quality of English Language

See above Comments and Suggestions for Authors. Errors in the new text were found. 

Author Response

Thank you for your valuable feedback and suggestions for improving the manuscript.

We sincerely apologize for any inconvenience caused by not highlighting the specific sections where corrections were made. We appreciate your understanding, and we will make it a priority to adhere to best practices in future manuscript revisions.

  1. None ‘Longitude’ changed to adjective ‘Longitudinal’ – Table 4
  2. Confusing ‘index’ delited in the line 472
  3. Thickness corrected in the table 4
  4. We found other spelling error in the line 281 ‘sufficient’ not ‘suficient’